# Effects of Silk Fibroin Enzyme Hydrolysates on Memory and Learning: A Review

**DOI:** 10.3390/molecules27175407

**Published:** 2022-08-24

**Authors:** Sidney J. Stohs, Luke R. Bucci

**Affiliations:** 1School of Pharmacy and Health Professions, Creighton University Medical Center, Omaha, NE 68178, USA; 2InnerPath Nutrition, Reno, NV 89523, USA

**Keywords:** silk fibroin enzyme hydrolysate (FEH), attention, learning, mental focus, memory retention, memory quotient, neuroprotection, antioxidant, clinical studies, mechanisms, safety, acetylcholine

## Abstract

Silk protein products have been used for a wide range of applications. This review focuses on the studies conducted relative to cognitive functions with silk fibroin enzyme hydrolysates (FEH) in humans and animals. All known studies reported in PubMed and Google Scholar have been included. Studies have been conducted on children, high school and college students, adults and seniors, ranging in ages from 7–92 years. Doses of 200–600 mg silk FEH per day for three weeks to 16 weeks have been used. Based on these studies, it can be concluded that silk FEH exhibit beneficial cognitive effects with respect to memory and learning, attention, mental focus, accuracy, memory recall, and overall memory and concentration. These conclusions are supported by studies in rats and mice. Mechanistic studies that have been conducted in animals and cell culture systems are also reviewed. These studies indicate that silk FEH exerts its positive effects on memory and learning by providing neuroprotection via a complex mechanism involving its potent antioxidant and inflammation-inhibiting activities. Acetylcholine (ACh) is secreted by cholinergic neurons, and plays a role in encoding new information. Silk FEH were shown to decrease the levels of the pro-oxidant and pro-inflammatory mediators interlukin-1 (IL-1β), IL-6 and tumor necrosis factor-alpha (TNF-α), protecting the cholinergic system from oxidative stress, thus enhancing ACh levels in the brain, which is known to promote cognitive functions. In addition, the expression of brain-derived neurotrophic factor (BNDF), which is involved in the survival of neurons, is enhanced, and an increase in the expression of the phosphorylated cAMP response element-binding protein (p-CREB) occurs, which is known to play a positive role in cognitive functions. No adverse effects have been reported in association with the use of silk FEH.

## 1. Introduction

In addition to the extensive use of silk protein in the textile industry, silk protein and silk protein hydrolysates have been widely consumed as food and as traditional medicines [1,2,3,4]. Silk-derived protein products have been reported to exhibit antidiabetic, antineoplastic, cardioprotective, neuroprotective, anti-hypercholesterolemic, wound healing, metabolic modulatory, and antioxidant properties [1,2]. They have also been used in tissue engineering, cosmetics and as carriers for drugs [1,2]. Furthermore, silk fibroin enzyme hydrolysates [FEH] have been shown to exhibit memory and learning enhancement, the subject of this review. Recently, in memory impaired rodents the memory enhancing and anti-Parkinson effects of a mature silkworm product prepared by steaming and freeze-drying has been demonstrated [5,6]. This product contains large amounts of fibroin, and the protective effects of the product may be due to the fibroin.

The cocoons of the silkworm moth, *Bombyx mori*, are composed of a single strand of raw silk which is up to 1500 m in length. It is composed of an outer protein shell called sericin and an inner double stranded core consisting of the protein fibroin [7,8,9]. The silk fibroin constitutes approximately 70% of the total silk protein. Silk fibroin is composed of the amino acids glycine (Gly), L-alanine (Ala), and L-serine (Ser) in the specific repeating sequence (Gly-Ser-Gly-Ala-Gly-Ala)_n_ in conjunction with smaller amounts of other amino acids [7,8,9]. The structure has also been represented as the repeating sequence (Gly-Ala-Gly-Ala-Gly-Ser)n [10].

The repeating amino acid sequence forms a common protein structure called beta sheets that are arranged to be adjacent when folded, offering great tensile strength [1,2,4].

Specifically prepared silk fibroin enzyme hydrolysates (FEH), with an average molecular weight of approximately 1500 daltons, have been shown to improve cognitive functions in normal, healthy humans [11,12,13,14,15,16,17,18,19,20]. These studies involving silk FEH have focused on various aspects of memory and learning functions and mental focus, and have included children, high school and college students, adults and seniors, ranging in ages from 7–92 years [11,12,13,14,15,16,17,18,19,20]. These studies, as well as various mechanistic studies in humans, animals and cell culture systems, are reviewed. A number of the studies were published in Korean language scientific journals for which there was no English translation available, and were translated into English using Google Translate.

## 2. Human Studies

A number of human as well as animal studies have assessed the ability of silk FEH to enhance various cognitive functions, including memory, learning, mental focus, concentration, reasoning, recall, accuracy, retention and memory quotient (a measure of overall memory and concentration) [11,12,13,14,15,16,17,18,19]. These studies are summarized in Table 1.

A human three-week placebo-controlled, double-blinded study was conducted involving 76 healthy subjects ranging in age from 28–92 years [11]. The subjects were randomly divided into four groups that received 0, 280, 400 or 600 mg silk FEH daily in two divided doses. Memory and learning were assessed at baseline and after 3 weeks using the Rey-Kim Auditory Verbal Learning Test and the Rey-Kim Complex Figure Test components of the Rey-Kim Memory Test (RKMT). After three weeks, dose-dependent increases in memory quotient, learning gradient, number of words remembered, retrieval efficiency, and drawing/recall scores were observed [11]. All were statistically significant (*p* < 0.05), with the exception of retrieval efficiency, which increased by approximately 40% but was not significant, due to large standard deviations. The optimal dose for silk FEH in this study was 400–600 mg per day, depending on the test.

The effects of an oral daily dose of 400 mg silk FEH for one month on memory and learning was assessed in 40 high school students using the RKMT [12]. The silk FEH was delivered orally in a product that contained DHA/EPA, which was also present in the placebo. After one month, memory preservation, memory index and attentive concentration were significantly improved in the silk FEH treated group relative to the control group, indicating a positive effect on learning memory.

Twenty-five male and female subjects, with an average age of 72 years, were randomized into two groups, a treatment group and a placebo group, with the treatment group receiving 200 mg silk FEW twice a day (400 mg/day) for three weeks [13]. The degree of mental functioning of the participants ranged from normal to severe dementia. Cognitive function was assessed using the Mini-Mental State Exam (MMSE), a widely employed test which measures memory registration, memory retrieval, attention, concentration, calculation, orientation, and spatial and temporal organization. A significant improvement in cognitive ability in the treatment group relative to the placebo was observed based on the overall MMSE test score. When treated subjects were separated, based on normal, mild dementia and severe dementia, greatest improvement in cognitive function was observed in the severe dementia group [13].

A placebo-controlled double-blinded cognitive function study was conducted in 66 normal subjects with an average age of 43 years (27–59 years) [14]. Subjects received placebo, 200 mg or 400 mg of a silk FEH orally daily. All subjects were assessed at baseline and after three weeks using the Wechsler Adult Intelligence Scale (WAIS) Test which provides a memory index score, as well as scores for verbal understanding, attention and concentration. A significant Silk FEH dose-dependent increase in overall cognition was demonstrated relative to the placebo control.

The effects of silk FEH on cognition involving 99 normal human subjects, ranging in age from 19 to 64, were reported in two publications [15,16]. The overall study was placebo-controlled, and double-blinded, with subjects daily receiving placebo, 200 mg or 400 mg Silk FEH for three weeks. Cognitive functions were assessed using the Rey-Kim Memory Test (RKMT) [15] and Rey Complex Figure Test [16]. The results demonstrated statistically significant dose-dependent increases in memory quotient, memory recall, number of words memorized and memory maintenance [15], as well as brain function in recognizing and memorizing complex two-dimensional figures [16]. Overall, the results provided evidence for the ability of silk FEH to improve memory and learning ability.

A placebo-controlled study was conducted in 46 normal children with an average age of about 10 years and IQ of about 116 [17]. The children were randomized into a placebo group and a treated group, with the treated students orally receiving 200 mg silk FEH twice daily (400 mg/day) for 16 weeks. The Tail Marking Test was used to assess cognitive function at baseline and endpoint. Silk FEH consumption significantly decreased response times and increased accuracy of performance relative to the control group, thus showing improvement in attention and cognitive flexibility. A deficiency of the study was the lack of testing at time points between baseline and the 16-week endpoint.

In another placebo-controlled double-blinded study assessing cognitive function in 36 male and female children with an average age of about 10 years, the subjects were randomly divided into a placebo group and a treated group, with the treated students receiving 200 mg silk FEW orally twice a day (400 mg/day) for four weeks [18]. Memory functions were tested using the RKMT for Children. As compared to baseline, statistically significant improvements were determined for immediate recall, delayed recall, copying, immediate figure recall, delayed figure recall, memory retention and retrieval efficiency.

Overall, the daily consumption of silk FEH by these children resulted in improvements in memory and learning. As with the previous study, the time-dependent effects of silk FEH consumption in these children were not determined, which would have added greatly to the knowledge base regarding the effects of this substance.

The association between cerebral blood flow and cognitive improvement was assessed in four healthy university students (two male and two female) with an average age of 23 years following daily administration of 400 mg silk FEH orally for 3 weeks [19]. Changes in memory and learning were determined at baseline and completion of the study using the WAIS test, while blood flow to the brain was determined by single photon emission computed tomography (SPECT). The results showed that following consumption of silk FEH, mean IQ increased from 103 to 114. In addition, blood flow to the brain increased in the para-hippocampal gyrus and medial temporal areas, suggesting that one of the mechanisms regarding the increased cognitive function associated with silk FEH might involve increased blood and oxygenation to the brain. Weaknesses of this study were the small number of subjects tested, the lack of a placebo group, and the need to be double-blinded [16]. The results are suggestive that increased blood flow to the brain may be involved in the mechanism of silk FEH, but the numbers of subjects and the design of the study do not allow this conclusion as a clear mechanistic result.

A placebo-controlled, double-blinded study involving 30 healthy male and female college students with an average age of about 21 years was reported in which students received either placebo or 200 mL milk containing 10 mg silk FEH daily for 30 days [20]. The students were tested at baseline and at termination of the study with two versions of the Paced Auditory Serial Addition Test (PASAT). The results indicated that daily consumption of the silk FEH increased accuracy with respect to working memory and attention, and mathematical ability, compared to baseline, when employing both versions of the test. The primary issue with this study relates to the very low daily dose that was used. Other studies giving silk FEH orally have shown that the threshold effective dose is about 200 mg per day in capsule form, as compared to 10 mg per day given in milk as used in this study, which raises serious questions regarding the results of this study [11,14,15,16].

In summary, various human studies have been conducted with silk FEH that demonstrate improvement in cognitive functions. Various tools and tests have been employed to assess various aspects of memory and focus.

## 3. Animal Studies

Various animal studies have assessed the ability of silk FEH to improve memory and cognition, affirming and extending efficacy results observed in humans. These studies are summarized in Table 2. Scopolamine acts by preventing the binding of acetylcholine to muscarinic receptors, as described below under Mechanism(s) of Action, and has been used as a tool to assess the effects of silk FEH on memory and learning. A study was conducted on Sprague-Dawley rats that were divided into three groups, control, scopolamine (1 mg/kg), and scopolamine plus silk FEH (10 mg/kg) [13]. The animals were subjected to two tests, a passive avoidance test and a water maze test. In both cases, scopolamine produced significant deficits in function as compared to controls, while silk FEH pretreatment prevented the deficits induced by scopolamine [13]. The authors concluded that the silk FEH improved memory and learning, as reflected by the results of the two tests.

The cognitive enhancement effects of various fractions of silk FEH were determined using an 8-arm maze test in scopolamine memory-impaired rats [25]. The results showed that a maximum of approximately 50% memory recovery (preservation) was achieved relative to control values with the mid (500–1000) molecular weight fraction while lesser protection was achieved with higher (1000–2000) and lower (<500) molecular weight fractions.

In another study involving the use of scopolamine to induce learning and memory impairment [22], groups of C57BL/6 mice were randomized and treated with scopolamine (1 mg/kg intraperitoneally), scopolamine plus silk FEH (10, 50 and 150 mg/kg orally for three weeks), scopolamine plus donepezil (2 mg/kg for three weeks), and a placebo control. Donepezil is a cholinesterase inhibitor that has been shown to prevent the progression of memory impairment and decline, and was used in this study as a positive control. The animals were subjected to a passive avoidance test, a Morris water maze test and a Y-maze test. Silk FEH, as well as the donepezil positive control, significantly suppressed scopolamine-induced memory and learning impairment, as demonstrated by all three tests, affirming the beneficial effects of silk FEH with respect to cognitive functions in this animal model. The 10 mg/kg daily FEH proved to be generally as effective as the 150 mg/kg dose [22].

Accumulation of the peptide amyloid-beta in the brain is known to attenuate cognitive functions [16,21]. In a study similar in design to the above study with scopolamine [22], a single intra-hippocampal injection of amyloid-beta (2 μg/3 μL) in C57BL/6 mice resulted in significant decrements in memory and learning, as determined by the Morris water maze test, a passive avoidance test and a Y-maze test [23]. Silk FEH (10, 50 and 150 mg/kg orally for three weeks) as well as donepezil (2 mg/kg for three weeks; positive control) resulted in significant protection from the deleterious effects of the amyloid-beta with respect to memory and learning [23]. The 10 mg/kg dose of FEH was as effective as the 150 mg/kg dose. Thus, pretreatment with silk FEH provided protection against the cognitive impairment produced by the amyloid-beta.

The cognition protection effects of silk FEH were assessed in Sprague-Dawley rats that were subjected to transient middle cerebral artery occlusion [15]. Following reperfusion to induce neuronal cell damage and memory impairment, the animals were treated with silk FEH (10 mg/kg) and assessed via a passive avoidance test, a rota-rod test and an 8-arm maze test relative to the sham control and vehicle control groups. Significantly fewer errors were observed in the amyloid-beta treated animals that also received silk FEH, relative to the animals that received only amyloid-beta. The results indicated neuroprotection and increases in memory and learning as a result of receiving silk FEH.

The ability of silk FEH to improve memory and learning has also been studied in rats subjected to focal cerebral ischemia [24]. Treatment of the animals with silk FEH (10 mg/kg) seven days before and seven days after inducing cerebral ischemia, indicated that silk FEH treatment significantly reduced the number of errors in an 8-arm maze test and increased latency time in a passive avoidance test. The results indicated that silk FEH treatment could improve ischemia-associated memory deficits.

In summary, in animal studies, impaired cognitive functions induced by scopolamine, amyloid-beta and brain ischemia were significantly improved by oral treatment with silk FEH, thereby providing supporting evidence for the efficacy of silk FEH.

## 4. Mechanism of Action

Several mechanisms have been proposed for the enhancement of cognitive functions by silk FEH, and various studies have demonstrated that FEH exerts multiple effects. As a consequence, complex effects are involved, which are summarized in Figure 1.

Various studies have demonstrated neuroprotective effects of silk FEH in humans, animals and cell culture systems which have been correlated with its antioxidant and anti-inflammatory activities [9,12,13,14,17,21,22,23,24,25,26]. The neuroprotective effects and cognitive enhancement, as well as other health-promoting effects, of silk FEH may be attributed to these potential effects that confer protection against free radicals and oxidative neuronal damage.

The brain has a high rate of oxygen consumption and is therefore very sensitive to oxidative stress. Although it is relatively small in size, the brain consumes approximately 20% of the total resting oxygen utilization by the body [14]. Damage due to oxidative stress occurs due to an imbalance between free radicals involved in physiological functionals and excess free radical formation. Due to its high demand for oxygen and its limited antioxidant capacity, the brain is exceedingly susceptible to an excess of reactive oxygen species, and subsequent oxidative stress and inflammation, which results in the disruption of the production of neurotransmitters and neurotransmission, culminating in neurological dysfunctions [3].

The mechanism(s) underlying the antioxidant properties of silk FEH may relate to its amino acid sequence, very high content of the hydroxyl amino acids, L-serine, L-tyrosine and L-threonine which chelate free radicals and reactive oxygen species (ROS), and, possibly, its pleated sheet structure [3,4,11].

If silk FEH acts as an antioxidant and neuroprotectant, how does the inhibition of oxidative stress and free radicals relate to the ability of this product to enhance memory and learning? Cognitive functions are closely associated with the cholinergic system, and brain levels of acetylcholine (ACh) are a biomarker of memory and learning [10,16,27,28]. Ach is secreted by cholinergic neurons, and plays a role in encoding new information. Other neurotransmitters, such as γ-aminobutyric acid (GABA), dopamine and serotonin, may also be involved. These neurotransmitters serve crucial roles as intercellular messengers in the nervous system [28,29].

Memory and learning impairment are associated with oxidative damage to the nervous system, especially to the cholinergic system [13,16,22,30]. The induction of neuro-inflammation is associated with memory and learning loss via neurodegeneration [31,32,33,34], and impairing cholinergic neurons with the inhibition of choline acetyltransferase activity for the synthesis of ACh results in memory loss and learning impairment [28,29,30]. Acetylcholinesterase (AChE) is the enzyme that inactivates Ach, and AChE inhibitors and anti-inflammatory drugs improve memory loss and learning [8,9,11].

The drug scopolamine inhibits memory and learning by inhibiting the binding of ACh to muscarinic receptors, and is, therefore, used as a model to study cognitive functions in animals [33,34]. In an initial study by Kim et al. [13], when rats were pre-treated orally with silk FEH prior to the administration of scopolamine, silk FEH significantly enhanced cognitive functions in the rats as compared to animals that only received scopolamine. The results provided evidence that silk FEH exerted its beneficial effects through an ACh-mediated mechanism.

Treating rats with amyloid-beta decreased brain ACh concentrations by 55% [16]. Amyloid-beta administration and accumulation in the brain attenuates memory and learning [16,23]. When the rats were concurrently treated with silk FEH, the brain ACh concentrations were restored to approximately 80 % of the control values, indicating at least partial protection, and that the enhancement of cognitive properties of silk FEH involves an increase in ACh levels in the brain [16].

The neuroprotective effects of silk FEH were assessed in amyloid-beta treated SK-N-SH human neuronal cells [21]. Silk FEH significantly attenuated amyloid-beta induced cell death (apoptosis). The authors looked at various parameters associated with apoptosis. An increase in the enzyme caspase-3 is associated with apoptosis. This enzyme was markedly increased in these cells by amyloid-beta, while pretreatment with silk FEH significantly attenuated the increases in this enzyme. A marked increase in generation of reactive oxygen species (ROS) in these cells occurred following treatment with amyloid-beta which was significantly attenuated by pretreatment of the cells with silk FEH [21].

Increased intracellular calcium is another marker associated with apoptosis. Amyloid-beta treatment resulted in a significant increase in intracellular calcium in these neuronal cells, while pretreatment with silk FEH significantly reduced this effect [21]. In addition, the amyloid-beta induced decrease in mitochondrial membrane potential, increase in cytochrome c release and caspase-9 activation, as well as nucleus condensation, and segmentation in the cells was significantly attenuated by pretreatment with silk FEH. Taken together, the results demonstrated the neuroprotective effects of silk FEH in this in vitro system.

In another study, oral silk FEH administration (10, 50 and 150 mg/kg per day for three weeks) was shown to significantly ameliorate amyloid-beta induced memory and learning impairment in mice [23]. Furthermore, silk FEH, as well as the positive control donepezil, treatment of mice that had received amyloid-beta resulted in an increase in brain-derived neurotrophic factor (BDNF) which promotes the survival of neurons. In amyloid-beta treated mice that received silk FEH or donepezil, concomitant reductions occurred in the pro-inflammatory cytokines, including interleukin-1β (IL-1β), interleukin-6 (IL-6) and tumor necrosis factor-α (TNF-α), all of which are associated with the production of oxidative stress and inflammation in brain tissues [23]. Furthermore, the amyloid-beta induced increase in acetylcholine esterase (AChE), the enzyme responsible for the inactivation of Ach, which was significantly reduced by FEH as well as the positive control donepezil. In general, similar results were obtained with doses of 10, 50 and 150 mg/kg silk FEH.

In this study, additional mechanistic information was obtained [23]. Silk FEH stimulated the BDNF/tropomyosin receptor kinase B/phosphatidylinositol-3-kinase/protein kinase B/mammalian target of rapamycin/postsynaptic density protein 95 pathway, affording neuronal protection. Furthermore, silk FEH suppressed the expression of proteins of the p75 apoptotic (programmed cell death) signaling pathway, including tumor necrosis receptor-associated factor 6, B-cell lymphoma 2 (Bcl2)/Bcl2-associated X protein, caspase-3, and nuclear factor-kappa B (NF-κB) in the brain tissues of mice that were treated with amyloid beta; thus offering additional protection, and further evidence of the antioxidant and anti-inflammatory properties of silk FEH.

In a further study, silk FEH treatment increased ACh and BDNF levels, and reduced the levels of pro-oxidant mediators IL-1β, TNF-α, and IL-6 in brain hippocampal tissue of mice with scopolamine-induced memory and learning impairment [22]. As previously noted, donepezil results in effects similar to silk FEH with respect to the prevention of scopolamine-induced deficits in memory and learning, providing additional evidence for the role of ACh in cognitive functions.

Silk FEH also decreased AChE levels in the hippocampus that were increased by scopolamine treatment alone [22,23]. Furthermore, FEH treatment of scopolamine-induced memory and learning impaired mice enhanced brain hippocampal BDNF and the expression of phosphorylated cAMP response element-binding protein (p-CREB), via the stimulation of the phosphatidylinositol-3-kinase/protein kinase B/mammalian target of rapamycin/postsynaptic density protein 95 (p-PI3K/p-AKT/mTOR/PSD95) pathway, and activation of extracellular signal-regulated kinases (ERK) and Ca^2+^/calmodulin-dependent protein kinase II (CaMKII). P-CREB has a well-known role in long term memory formation and learning [35,36].

It has been suggested that the beta sheet portions of FEH may be involved in the mechanism of action of silk FEH [11]. This involvement could occur either through the antioxidant activities of the hydroxyl amino acids or via binding to beta sheet portions of other proteins, such as amyloid-beta, thereby preventing aggregation of amyloid beta which impairs memory and learning [12,19,37,38,39,40].

As noted above, a study was conducted which assessed cerebral blood flow and cognition in four healthy university students (two male and two female) following administration of 400 mg silk FEH orally for 3 weeks [19]. Blood blow increased in the brain para-hippocampal gyrus and medial temporal areas, suggesting that increased cognitive function associated with silk FEH consumption might involve increased blood and oxygenation to the brain. However, drawing this conclusion is tenuous because a small number of subjects were tested, and the study was not placebo-controlled and double-blinded.

The ability of various fractions of silk FEH were assessed for their protection against hydrogen peroxide-induced cell death in PC12 neuronal cells [25]. Greatest protection was observed with the low (<500) molecular weight fraction with decreasing protection by the mid (500–1000) and high (1000–2000) molecular fractions [20,25]. The authors provided no explanation for the differences in the results between this in vitro study and their in vivo study, in which the various fractions were tested in an 8-arm maze test in scopolamine-treated rats [20,25]. In the in vivo study, greatest cognitive protection was afforded by the mid-molecular weight fraction. However, the highest protection against hydrogen peroxide-induced cell death may have been due to the availability of a greater number of hydroxyl groups with low molecular weight fraction, as compared to the longer polypeptides.

In summary, these results indicate that silk FEH may exert its positive effects on memory and learning by offering neuroprotection via a mechanism involving its potent antioxidant and anti-inflammatory activities with protection of the cholinergic system. These actions may result in an inhibition of the breakdown of Ach with enhancement of synaptic levels, the enhanced expression of BNDF, which is involved in the survival of neurons, and the enhanced expression of p-CREB, which is known to play a positive role in cognitive functions.

## 5. Safety of Silk FEH

Silk proteins have been used as a food, food additive and traditional medicine for many years [1,2,3,4,5,6]. Silk protein powders have FDA Generally Recognized as Safe (GRAS) designation [41]. This determination is based at least in part on use of other proteins in food, as food additives or in food supplements. Silk protein powder is prepared by a hydrolysis method [41]. The GRAS notification states that silk protein powder “may be consumed in most any amount” but also suggests levels based on different purposes or needs. “The usual level is 2–5 grams or a flat teaspoon (5 grams). For fitness and intense activity and for muscle tone or ‘look good, feel good’ muscle, 30 grams is suggested on a daily basis”. This powder may be used in or on any food or beverage [41]. It should also be noted that FEH has been manufactured and sold as a healthy functional food in the Republic of Korea since 2009 [42].

A recent detailed toxicological assessment of silk fibroin was conducted in male and female Sprague-Dawley rats [43]. The studies demonstrated a very high degree of safety, with the data raising no questions regarding toxicological, mutagenic, genotoxic or allergenic concerns. An initial 14-day dose range finding study in rats was conducted with doses up to 500 mg/kg/day. Higher doses could not be used, due to solubility and oral gavage volume constraints. No silk fibroin dependent effects were observed.

This study was followed by a 28-day study in rats of both sexes to assess systemic toxicity on silk fibroin at daily doses of 0, 125, 250 or 500 mg/kg body weight [43]. No significant effects were observed with respect to body weight gain, food consumption, urinalysis, serum chemistries, hematology, visual effects or behavior. Due to the lack of clinical observations the no-observed adverse effect level (NOAEL) and the medial lethal dose (LD50) were determined to be greater than, or equal to, 500 mg/kg/day, the highest test dose. A bacterial mutation test (Ames’ test) with five bacterial strains revealed no mutagenic effects. An in vitro pepsin digestion assay assessed the potential for protein allergy, and produced negative results. In a mouse study, silk fibroin did not result in an increase in the average percentage of micro-nucleated reticulocytes. Thus, silk fibroin under these conditions was shown to be non-toxic and non-allergenic.

A single dose toxicity study in male and female Sprague-Dawley rats demonstrated that silk FEH was non-toxic (Prof. DK Kim, unpublished). Rats were given a single dose of 0, 250 mg/kg, 1.0 g/kg or 4.0 g/kg of silk FEH in phosphate buffered saline. After 14 days there were no significant changes in body weights, organ weights, clinical observations, or liver function tests. All animals survived and there were no abnormal signs or symptoms in any of the animals. The NOAEL was equal to, or greater than, 4.0 g/kg body weight. The highest dose used in these rats was over 700 times greater than a 400 mg dose in humans, a very wide margin of safety.

The Expert Panel for Cosmetic Ingredient Safety reviewed the safety of silk protein ingredients [44]. The panel concluded that fibroin and hydrolyzed fibroin were safe for inclusion into cosmetics based on present use and concentrations. Finally, it should be noted that no adverse events or reactions of any kind were reported in any of the human studies that have been conducted following oral administration of silk FEH at doses of 200–400 mg/day for 3–16 weeks [11,12,13,14,15,16,17,18,19,20]. Taken together, the above data indicates that silk FEH has an exceedingly high degree of safety.

## 6. Discussion

A series of human studies [11,12,13,14,15,16,17,18] have shown that silk fibroin enzyme hydrolysates (FEH) can enhance cognitive functions. Several dose-response effect studies have been conducted, with doses ranging from 200–600 mg silk FEH per day. The threshold dose appears to be about 200 mg per day [14,15,16], with doses of 400 mg/day generally providing maximal effects.

Various tests were used to assess cognitive functions in these studies, including the Rey-Kim Memory Test (RKMT), Rey-Kim Auditory Verbal Learning Test, the Rey-Kim Complex Figure Test components of the RKMT, the Tail Marking Test, the RKMT for Children, the Mini-Mental State Exam (MMSE), and Wechsler Adult Intelligence Scale (WAIS) Test. As a consequence, the positive cognitive effects of silk FEH were affirmed by a variety of widely used and adjudicated tests.

Studies were conducted on subjects ranging in age from 7–92 years [11,12,13,14,15,16,17,18], with positive results at all ages. In one study involving 76 subjects, the subjects ranged in age from 28–92 years [11]. Since the subjects were divided into four randomly assigned dosing groups, no conclusions could be drawn with respect to any age group, due to an insufficient number of subjects. However, in general, positive cognitive effects were reported in various studies at all ages. Furthermore, studies demonstrating positive cognitive effects were reported for groups of individuals with average ages of 10 years [17,18], 17 years [12], 42–43 years [14,15,16], and 72 years [13].

A weakness of the human studies that have been conducted with silk FEH is the lack of ethnic diversity among the test subjects. A need exists for studies that involve much more broadly based ethnic groups. Such results would add substantiation of the currently observed results to the general population.

Human studies involving silk FEH have ranged in duration from three to 16 weeks. However, no time course studies have been conducted, and, therefore, it is not clear whether the effects seen at three or four weeks are sustained at longer time points, and at what time point maximal effects are observed. Such studies should be conducted at various doses.

A number of animal studies have been conducted which provide supporting data with respect to the beneficial effects of silk FEH on memory and learning [13,14,15,21,22,23,24,25,26]. Silk FEH was shown to significantly inhibit the adverse effects of agents such as scopolamine and amyloid-beta, as well as cerebral ischemia, on cognitive impairment. The assessment protocols used to assess memory and learning involved the Morris water maze test, a Y-maze test, and a passive avoidance test. The animal studies provided useful information in support of the human studies regarding memory and learning potential of FEH, as well as mechanistic information which cannot be ascertained in humans. However, no animal studies have been conducted in healthy young animals or aged animals as compared to human studies that have been conducted in humans over a wide age range.

With respect to the mechanism(s) of action, numerous animal and in vitro studies have demonstrated a possible role of oxidative stress and inflammation in cognitive impairment with silk FEH being able to provide significant neuroprotection [12,13,14,15,20,22]. Scopolamine, amyloid-beta and cerebral ischemia have been used to impair memory and learning in rats and mice, and these adverse effects were, at least in part, reversed by silk FEH administration.

Mitogen activated protein-kinase (MAPK) is an essential element in what is sometimes referred to as the inflammatory cascade, the sequence of events that result in production of reactive oxygen species and oxidative stress, neuro-inflammation, neuronal cell death, and memory impairment and decline [45,46,47]. P38 MAPK is activated by amyloid-beta. Incubation of human neuroblastoma cells and rat brain hippocampal neurons with amyloid-beta results in significant increases in MAPK and cell deaths as well as cognitive decline and memory loss [26]. When silk FEH is added, significant decreases in MAPK occur, resulting in neuroprotection and providing evidence for the role of MAPK in the neuroprotective effects of silk FEH.

The results suggest that silk FEH may exert its positive effects on memory and learning by protecting the cholinergic system, preventing the breakdown of Ach with enhancement of synaptic levels, and enhancing expression of BDNF, which is involved in the survival of neurons, as well as enhancing expression of p-CREB, which plays a positive role in cognitive functions. Based on these studies, a complex series of events appears to be involved in the neuroprotective effects of silk FEH. The degree to which the unique beta sheet structure impacts the mechanism of action of silk FEH has not been fully elucidated.

Taken together, the results provide evidence for the cognitive enhancing effects of silk FEH, involving a mechanism related to its antioxidant and inflammation-inhibiting effects with protection and enhancement of the cholinergic system with no demonstrated adverse effects.

## 7. Conclusions

Studies in humans ranging in ages from 7–92 years at doses of 200–600 mg silk FEH per day for three weeks to 16 weeks have demonstrated beneficial cognitive effects. It can be concluded that silk FEH exhibit beneficial effects with respect to memory and learning, overall memory and concentration, accuracy, memory recall, attention and mental focus. These conclusions based on human studies are supported by studies in rats and mice. Mechanistic studies in animals and cell culture systems indicate that silk FEH exerts its positive effects on memory and learning by providing neuroprotection via a complex mechanism involving its potent antioxidant and inflammation-inhibiting activities. Human, animal and cell culture studies indicate that silk FEH has an exceedingly high degree of safety.

## Figures and Tables

**Figure 1 molecules-27-05407-f001:**
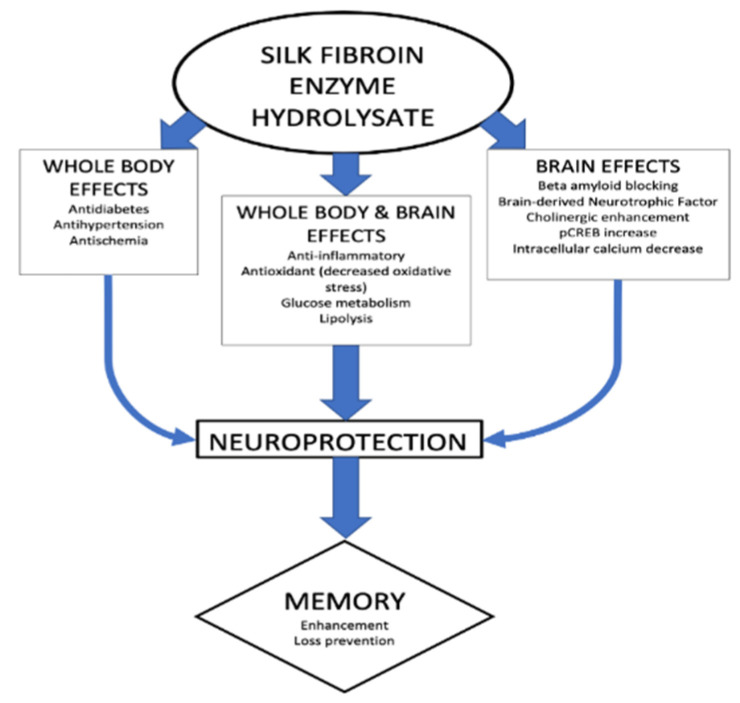
Multiple mechanisms for effects on memory of silk fibroin enzyme hydrolysates.

**Table 1 molecules-27-05407-t001:** Silk Fibroin Enzyme Hydrolysate Clinical Human Studies.

Reference	# of Subjects	Age Range	Duration (Weeks)	Daily Dose (mg)	Significant Memory Enhancement Effect
[11]	76	28–92	3	0, 280, 400, 600	learning, recall
[12]	40	6–19	4	0, 400	attention, memory preservation
[13]	25	70–82	3	0, 200 × 2	0verall cognition
[14]	66	27–59	3	0, 200, 400	Overall cognition
[15]	99	16–64	4	0, 200, 400	recall, memory quotient
[16]	99	16–64	4	0, 200, 400	memory quotient
[17]	46	9–11	16	200 × 2	attention, cognition
[18]	36	7–12	4	0, 200 × 2	recall, attention

**Table 2 molecules-27-05407-t002:** Silk Fibroin Enzyme Hydrolysate Animal Studies.

Reference	Animals	Memory/Learning Impairment	Test	Memory Preservation by FEH (% of Control)
[13]	rats	scopolamine	passive avoidance	62
	rats	scopolamine	water maze	60
[21]	rats	scopolamine	8-arm maze	50
[22]	mice	scopolamine	passive avoidance	86
	mice	scopolamine	water maze	95
	mice	scopolamine	Y-maze	90
[23]	mice	amyloid-beta	passive avoidance	69
	mice	amyloid-beta	water maze	60
	mice	amyloid-beta	Y-maze	81
[15]	rats	artery occlusion	8-arm maze	79
[24]	rats	artery occlusion	8-arm maze	63
	Rats	artery occlusion	passive avoidance	76

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
