# Peer review of "Effects of Silk Fibroin Enzyme Hydrolysates on Memory and Learning: A Review"

_molecules, 2022, doi:10.3390/molecules27175407_

Round 1

Reviewer 1 Report

The authors review the literature on the effect of silk fibroin enzyme hydrolysates on cognitive function and brain health. There seems to be a somewhat broad enough scope of studies on FEH to warrant a review.

Most of the human studies were published in Korean. Can one of the authors read Korean? Do they have a translator? Did they run the text through Google Translate or the like?  Please indicate it the footnotes at the end.

The design of most if not all of the human studies are "before vs after". They seem to rely on improvement in test performance after treatment compared to performance before. Is this necessary for these tests? Couldn't a simple between-subjects design be used? A within-subjects design may allow for practice effects. The authors should discuss this similar design feature among the human studies, and possible issues and concerns attached to it.

All of the rodent studies involve inducing some sort of impairment before FEH treatment, such as by scopolamine, a-beta, or cerebral ischemia, while most of the human studies are in healthy young individuals (one exception), who are not likely to experience such impairments. However, they may instead improve in ability over time, as children do. Are there any examples of FEH improving performance in healthy young or even aged animals? If not, then the authors should discuss these differences between human and animal studies, and any issues it brings up about humans use of FEH.

It would be helpful if the authors brought up questions that have yet to be answered within this line of research. This may make this review more useful to the reader.

The sentence in line 116 seems grammatically odd. It seems to have a subject, but not a predicate.

Line 82, FEW--> FEH (?)

Line 110, Tail Marking Test -->Color Trails Making Test. The name of this test is mentioned in the English version of this study's abstract, and the authors should not have missed it. It raises concerns about how thoroughly the human literature was actually reviewed.

Author Response

Dear Reviewer of our review manuscript entitled “Effects of Silk fibroin Enzyme Hydrolysate on Memory and Learning:  A Review”.  We sincerely appreciate the helpful suggestions concerning our review. Your queries will strengthen our review and eliminate any ambiguities and errors.  Below are responses to your queries which we have noted in red type below each.

Most of the human studies were published in Korean. Can one of the authors read Korean? Do they have a translator? Did they run the text through Google Translate or the like?  Please indicate it the footnotes at the end.

Response: We have noted at the end of the Introduction that a number of the studies were published in Korean language scientific journals for which there was no English translation available, and were translated into English using Google Translate. We did in fact use Google Translate, although for one paper, we had a Korean individual translate selected sections and compared with the results for Google Translate.  The translations were very comparable, and we were satisfied with the results.

The design of most if not all of the human studies are "before vs after". They seem to rely on improvement in test performance after treatment compared to performance before. Is this necessary for these tests? Couldn't a simple between-subjects design be used? A within-subjects design may allow for practice effects. The authors should discuss this similar design feature among the human studies, and possible issues and concerns attached to it.

Response:  The various dosing regimens are noted in Table #1, and all but one of the studies employed a 0 dose (control) group.  As a consequence, we do not believe that this is an issue.  Results were always compared between groups receiving FEH and the 0 dose (control) group for these studies.

All of the rodent studies involve inducing some sort of impairment before FEH treatment, such as by scopolamine, a-beta, or cerebral ischemia, while most of the human studies are in healthy young individuals (one exception), who are not likely to experience such impairments. However, they may instead improve in ability over time, as children do. Are there any examples of FEH improving performance in healthy young or even aged animals? If not, then the authors should discuss these differences between human and animal studies, and any issues it brings up about human use of FEH.

Response:  As summarized in table 1, the studies used individuals ranging in age from 7-92, with only three studies using individuals below the age of 20. The average age for the other 5 studies ranged from 42-72.  As a consequence, improvements were observed at all ages. We have noted in the Discussion that the animal studies provide useful information in support of the human studies regarding memory and learning potential of FEH, as well as mechanistic information which cannot be ascertained in humans.  However, no animal studies have been conducted in healthy young animals or aged animals as compared to human studies that have been conducted in humans over a wide age range.

It would be helpful if the authors brought up questions that have yet to be answered within this line of research. This may make this review more useful to the reader.

Response:  In the discussion, we have noted that all studies to date have been in Asians, and studies in non-Asians are needed.  We have also discussed the fact the no long term studies have been conducted with determinations at various time points.  These should also be done at various doses.  It would also e useful to determine oxidant/antioxidant blood levels using various techniques over time and at various doses.

The sentence in line 116 seems grammatically odd. It seems to have a subject, but not a predicate.

Response: ”which” has been replaced with “the subjects” such that the sentence now reads more clearly.

Line 82, FEW--> FEH (?)

Response:  Correction made

Line 110, Tail Marking Test -->Color Trails Making Test. The name of this test is mentioned in the English version of this study's abstract, and the authors should not have missed it. It raises concerns about how thoroughly the human literature was actually reviewed.

Response:  This typographical error was not caught by the authors and has now been corrected.

Reviewer 2 Report

This manuscript is an interesting review on the improvement of memory and learning ability, which are the revealed functionalities of fibroin enzymatic hydrolysates. Fibroin is the major protein of Silk fibers. Currently, dementia and memory loss are major social and national problems worldwide, so I considered it a timely review. However, there are major comments that need to be corrected before the paper is published.

Major comments

 1.    The authors mentioned (Gly-Ser-Gly-Ala-Gly-Ala)n as the peptide repeat constituting Fibroin. However, Mita et al. (1994) reported Gly-Ala-Gly-Ala-Gly-Ser as repetitive sequences. Thus, in addition to the papers cited in this manuscript, it is necessary to refer to various papers that have researched the structure of fibroin to provide more accurate information regarding repetitive amino acid sequences in Fibroin.

Mita et al., (1994) Highly repetitive structure and its organization of the silk fbroin gene. J. Mol. Evol.38: 583-592.

2.  1.    Since this review is about the functionality of silk fibroin from silk moths, it is necessary to mention products with memory-promoting effects among silkworm products other than fibroin enzymatic hydrolysate (FEH). For example, recently, in the case of Hongjam, a product manufactured for human consumption by steaming and freeze-drying mature silkworms which contain enlarged silk glands with silk fiber protein, the effect of improving the nervous system functions such as improving memory and preventing the onset of Parkinson's disease has been reported. Hongjam is a product that contains large amounts of silk proteins including Fibroin, and the protective effect on the nervous system and the prevention of the onset of Parkinson's disease is most likely due to Fibroin. Therefore, it is necessary to refer to the papers on the functionality of Hongjam and to compare the functionality between Hongjam and FEH.

Park et al., (2022) Sericulture and the edibleinsect industry can help humanity survive: insects are more than just bugs, food, or feed. Food Sci. Biotechnol. 31(6) 657-668.                     

3. Organizing the animal and clinical test results in separate tables will help readers quickly understand the research results. 

4.  It is necessary to explain the mode of action of FEH with a schematic model.

5. In the Safety of Silk section, it is necessary to mention that FEH has been manufactured and sold as a healthy functional food in Republic of Korea since 2009.

Once, the suggestions are corrected, the publication of this review will be possible.

Author Response

Dear Reviewer of our review manuscript entitled “Effects of Silk fibroin Enzyme Hydrolysate on Memory and Learning:  A Review”.  We sincerely appreciate the helpful suggestions concerning our review, your queries will strengthen our review and eliminate any ambiguities and errors.  Below are responses to your queries which we have noted in red type below each.  We believe that we have appropriately addressed all of the queries.

  1. The authors mentioned (Gly-Ser-Gly-Ala-Gly-Ala)n as the peptide repeat constituting Fibroin. However, Mita et al. (1994) reported Gly-Ala-Gly-Ala-Gly-Ser as repetitive sequences. Thus, in addition to the papers cited in this manuscript, it is necessary to refer to various papers that have researched the structure of fibroin to provide more accurate information regarding repetitive amino acid sequences in Fibroin.

Response:  We have noted in the Introduction that he structure has also been represented as the repeating sequence (Gly-Ala-Gly-Ala-Gly-Ser)n and have cited the Mita et al reference. We believe that the two sequences are essentially the same, and differ only in where one makes the repeating cut.

  1. Since this review is about the functionality of silk fibroin from silk moths, it is necessary to mention products with memory-promoting effects among silkworm products other than fibroin enzymatic hydrolysate (FEH). For example, recently, in the case of Hongjam, a product manufactured for human consumption by steaming and freeze-drying mature silkworms which contain enlarged silk glands with silk fiber protein, the effect of improving the nervous system functions such as improving memory and preventing the onset of Parkinson's disease has been reported. Hongjam is a product that contains large amounts of silk proteins including Fibroin, and the protective effect on the nervous system and the prevention of the onset of Parkinson's disease is most likely due to Fibroin. Therefore, it is necessary to refer to the papers on the functionality of Hongjam and to compare the functionality between Hongjam and FEH.

Response:  At the end of the first paragraph of the introduction we have noted that” the memory enhancing and anti-Parkinson effects of a mature silkworm product prepared by steaming and freeze-drying has been demonstrated. This product contains large amounts of fibroin, and the protective effects of the product may be due to the fibroin.”  Several references have been cited.

  1. Organizing the animal and clinical test results in separate tables will help readers quickly understand the research results. 

Response:  We have included two tables in the manuscript, Table #1 which summarizes human studies, and Table #2 which summarizes animal studies.

  1. It is necessary to explain the mode of action of FEH with a schematic model.

Response:  Figure #1 has been included which summarizes the rather complex and multifaceted mechanism of action.

  1. In the Safety of Silk section, it is necessary to mention that FEH has been manufactured and sold as a healthy functional food in Republic of Korea since 2009.

Response:  We have noted at the end of the first paragraph under Safety that that FEH has been manufactured and sold as a healthy functional food in the Republic of Korea since 2009.

Round 2

Reviewer 1 Report

The addition of the tables is useful. Not sure what Table 2's "memory recovery" is exactly. It may help to explain how this was calculated.

Author Response

 We again thank you for your helpful comment which indicated that you were not sure what Table 2's "memory recovery" is exactly.

Response: Yeo et al (2004) used the term “memory recovery” to reflect the ability of FEH to prevent scopolamine induced memory loss relative to control animals.  We have changed the term to “memory preservation” since FEH can be viewed as preventing memory loss due to scopolamine.  In the table we have also noted that percent reflects percent relative to control values.

Reviewer 2 Report

The revised version of the manuscript has improved a lot. There are some typos and missing references. After changing, it is ready to publish. 

Comment 1. Line 40 Recently, the memory enhancing and anti-Parkinson effects of a mature silkworm product prepared by steaming and freeze-drying has been demonstrated [5,6]. 

Please add "memory enhancement effects in mild cognitive impairment rodent models" within this sentence. 

Comment 2. Line 66. Please remove 3 in the middle of a sentence. 

Comment 3. Line 395 ~ 397. Recent reviews by Park et al. 2022 summarized healthy functional foods from silkworms currently available in Korea. Please cite this article for an appropriate reference.

Park et al., (2022) Sericulture and the edible‑insect industry can help humanity survive: insects are more than just bugs, food, or feed. Food Sci. Biotechnol. 31(6) 657-668.

Author Response

We again thank you for your helpful comments.  We believe that we have addressed all of your queries and made the appropriate changes in the manuscript.  Point by point responses are found below.

The revised version of the manuscript has improved a lot. There are some typos and missing references. After changing, it is ready to publish. 

Response: We have conducted a spell check and corrected several errors.  W have added the reference noted below, as well as added reference numbers as appropriate.

Comment 1. Line 40 Recently, the memory enhancing and anti-Parkinson effects of a mature silkworm product prepared by steaming and freeze-drying has been demonstrated [5,6]. 

Please add "memory enhancement effects in mild cognitive impairment rodent models" within this sentence. 

Response:  Have incorporated this phrase into the appropriate sentence. 

Comment 2. Line 66. Please remove 3 in the middle of a sentence. 

Response:  the “3” has been deleted. 

Comment 3. Line 395 ~ 397. Recent reviews by Park et al. 2022 summarized healthy functional foods from silkworms currently available in Korea. Please cite this article for an appropriate reference.

Park et al., (2022) Sericulture and the edible‑insect industry can help humanity survive: insects are more than just bugs, food, or feed. Food Sci. Biotechnol. 31(6) 657-668.

Response:  This reference has been included, and the reference numbers corrected.